# Factors associated with adherence to antiretroviral therapy among HIV-positive adolescents and young adult patients attending HIV care and treatment clinic at Bombo Hospital in Tanga region-Tanzania

**Sophia Kamote**\*, **Novatus Apolinary Tesha**◉\*, **Bruno F. Sunguya**

School of Public Health and Social Sciences, Muhimbili University of Health and Allied Sciences, Dar es Salaam, Tanzania

\* sophykamote@yahoo.com (SK); novatus.tesha@muhas.ac.tz (NAT)

## Abstract

### Background

Adherence to HIV treatment regimens involves the consistent and correct intake of all prescribed medications. The implementation of antiretroviral therapy (ART) program has significantly reduced mortality among adolescents living with HIV. However, adherence to ART is lower among adolescents compared to other sub-populations and even lower in sub-Saharan Africa. The factors influencing ART adherence are context-specific and vary across countries and regions. In the Tanzanian context, there is a paucity of data regarding these factors.

### Methodology

This cross-sectional study involved 385 adolescents and young adults living with HIV receiving treatment at Bombo Hospital Referral Hospital's Care and Treatment Clinic, in Tanga, Tanzania. To assess adherence, a one-month self-recall medication adherence scale was used while a structured questionnaire was used to gather data on determinants of adherence. Data were collected using Google Forms and subsequently exported as a Microsoft Excel file. The data were then entered into Stata software version 15 for cleaning for descriptive and logistic regression analyses.

### Results

More than a third (35.3%) of adolescents and young adults living with HIV in Tanga were not adherent to the effective and available ART. Adolescents and young adults living in households experiencing moderate food insecurity were 67% less likely to adhere to ART (95%CI 0.16–0.66) compared to those who were food secure. Those with secondary education were 2.3 times more likely to adhere to ART (95%CI 1.02–5.23), compared to those without formal education. While participants who consistently obtain their ART at the clinic were

**Data Availability Statement:** Data pertinent to this study are stored at the Muhimbili University of

Health and Allied Sciences' repository. The data is owned by Muhimbili University of Health and Allied Sciences Ethical Clearance Committee. It may be available to share upon request to the Director of Research and Publications, Muhimbili University of Health and Allied Science, P.0. Box 65001, email: drp@muhas.ac.tz.

**Funding:** The author(s) received no specific funding for this work.

**Competing interests:** The authors have declared that no competing interests exist.

more 4.2 times more likely to adhere to medication (95%CI 1.29–13.72), those experiencing ART side effects were 39% less likely to adhere to ART (95%CI 0.38–0.98).

## Conclusion

More than one-third of adolescents and young adults were not adherent to ART in Tanga, Tanzania. Addressing such unprecedented challenges calls for efforts targeting adolescents and young adults with limited education, from households with food insecurity, and ensuring counseling and management of ART side effects.

## Introduction

HIV/AIDS is a growing public health problem among adolescents both in morbidity and mortality, escalated by a surge of new infections, and low identification coverage. Elimination of HIV by 2030 called for fast track targets by the Joint United Nations Program on HIV/AIDS (UNAIDS) [1]. These 95-95-95 targets were meant to ensure 95% of all people living with HIV know their HIV status, 95% of those who are aware of their status access Antiretroviral therapy (ART), and 95% of those accessing ART are virally suppressed [1–3]. Adolescents and young adults are lagging behind in all these targets globally, jeopardizing the overall global target. To the contrary, new infections and deaths are increasing in this vulnerable population like in other key populations. In 2020, a total of 410,000 of those aged 10–24 years old were newly infected [1,2]. New HIV infections can be controlled when populations of the HIV infected individuals have their HIV viral loads controlled under Antiretroviral therapy (ART). The phenomenon commonly known as undetectable equals untransmittable [3,4].

ART has significantly reduced mortality rates among adolescents living with HIV, yet maintaining adherence to ART continues to be a considerable challenge [1,2,5]. By December 2022, 76% of all people living with HIV were accessing ART, with 77% of these individuals being adults aged 15 years and older, and only 57% being children aged 0–14 years [1,3]. Only 62.3% of [6–8] adolescents in sub-Saharan African countries have self-reported being adherent to ART [4,9]. Maintaining ART adherence above 95% is critical for effective viral suppression, preventing ART drug resistance, and extending life expectancy [5]. ART provides numerous clinical benefits, including viral suppression, disease progression halting, immune system restoration, improved survival rates, reduced morbidity, and enhanced quality of life [6,7].

Counseling, incentives, mobile phone short message service (SMS), peer-delivered behavioral intervention, community ART delivery, electronic adherence service monitoring device, lay health worker lead group intervention, and food assistance are effective in increasing HIV treatment uptake [7,10,11]. These interventions are either effective individually or when combined. Adolescent ART adherence has been associated with demographic, socioeconomic, and behavior-related factors among others [9,12]. Evidence varies with the context due to differences in cultural, economic, and community dynamics, making it necessary to consider the local context of each study [6,8]. In Tanzania, evidence is limited to a study done in its business capital [2], where the context is different from that of other regions owing to differences in the burden of the diseases, new infections rates, cultural and beliefs dynamics, and socio-economic disadvantages of the adolescents and young adults. The current study therefore, was conducted at Bombo Hospital in Tanga, Tanzania to assess the ART adherence and identify its associated factors among the population receiving ART at the hospital.

## Methodology

### Study design and setting

This hospital-based cross-sectional study used a quantitative approach to investigate the magnitude and factors associated with adherence among adolescents and young adult clients who received treatment at the Care and Treatment Clinic (CTC) of Bombo Hospital. This referral regional hospital serves the entire population of the Tanga region, where HIV is prevalent among 5.4% of adults aged 15–49 years of age, and 0.8% among those aged 0 to 14 years [13]. Tanga has 603 health facilities dedicated to providing HIV/AIDS care and treatment. In addition to the one regional referral hospital, the region has other 10 district hospitals, 38 public health centers, 366 public dispensaries, and 188 private health facilities. Notably, Bombo Regional Referral Hospital is central to referred cases within the Tanga region, ensuring that patients receive comprehensive and specialized care when needed [13].

### Study populations

The study population was adolescents aged 15–19 years and young adults aged 19–24 years living with HIV and on ART for at least 6 months registered at Bombo CTC during the three months data collection period. We excluded adolescents and young adults who were too ill to participate.

### Sample size

The sample size for the study was calculated using Cochran's formula.

$$N = \frac{Z^2 \times P(1-P)}{e^2}$$

Z is the level of confidence (1.96); P is the proportion of reported adherence to ARVs among young adults (50%), and e is the absolute margin of error (5%). Therefore, the minimum required sample size was 384.

### Sampling methods

The Bombo Regional Hospital was purposively selected. Participants were selected through a systematic sampling procedure, where study units were selected from the ART register using a pre-calculated sampling interval. The ART register was used as the sampling frame, and an interval of five was used. For participant recruitment, every fifth registered client was selected to be part of the study.

### Variables and measures

Adherence to ART was the dependent variable. It was assessed using the internationally validated Morisky Medication Adherence Scale (MMAS) one-month self-recall medication adherence scale, which included eight items with a binary response of Yes or No. Questions items asked the participant to answer YES or NO to items numbered one (1) through seven (7), with the eighth item having a five (5)-point Likert scale response. An overall higher score of six and above (6) to eight indicated that a participant was adherent to ART, and a score of less than six (6) to zero indicated that a participant was non-adherent to ART [14,15].

Independent variables included individual factors such as age, education level, gender, family size, employment status, and marital status. They were adopted from the Demographic Health Survey (DHS) questionnaire [16]. Age was collected as the number of years lived. The

educational status of the participants was categorized as completed primary school, post-primary school training, secondary ordinary level, secondary advanced level, post-secondary advanced level training, college, or university and this provided insight into how education influences health literacy and adherence to ART. Gender was categorized as female and male, which explored potential differences in ART adherence patterns between genders. Family size was determined by the number of family members residing in the household and explained how family support or burden impacts adherence. Current marital status was categorized as single, married, cohabiting, divorced, separated, or widowed. Employment status was determined by asking participants to indicate whether they were self-employed, government-employed, government temporary or contract employees or private permanent or contract employees and this explained the economic factors influencing adherence.

The tool for assessing drug-related factors was adopted from a study conducted in Tabora Tanzania [5]. Participants were required to indicate the number of pills they took per day. Those who reported taking more than two pills were categorized as having a high burden of pills. Participants were also asked about the side effects of their ART drugs, which was categorized as either experienced or not experienced side effects. Furthermore, the frequency of tabs taken per day was categorized as once, twice, or more and this provides insights on the complexity or inconvenience of ART dosage on the adherence.

The respondents were asked if they had experienced a shortage of ART drugs in the past and if they were provided with regular health education which assessed the reliability of drug supply and its impact on adherence. Waiting time among study participants was categorized as less than 1 hour, 1 to 5 hours and more than 5 hours which assessed its impact on ART adherence and distance between the respondent's home and the health facility was recorded. These factors were assessed as in a previous study on assessing ART adherence among adolescents conducted in Tabora [5].

The wealth index was calculated using household assets obtained from a questionnaire adapted from the DHS [16]. Principle Component Analysis (PCA) was used to reduce dichotomized variables to those loaded as factor one, which would better describe wealth index variations. Weighted wealth was determined by summing up the factor loading and divided into quintiles.

Food security was measured using the Household Food Insecurity Access Score (HFIAS) tool which has a total score of 27 and comprises 9 questions [17]. The scale was used to categorize households into food secure, mildly insecure, moderately insecure, and severely insecure. This assessed how the presence or absence of food can impact ART adherence. HFIAS had a Cronbach's alpha coefficient of 0.914.

## Data collection

Participants were interviewed in a private room at the CTC within the hospital. They were allowed to voluntarily consent to participate in the study and had the right to choose whether to disclose information or withdraw from the study at any point during data collection. The data collection exercise took place between 23$^{rd}$ January to 28th April 2023.

## Ethical consideration

The study received ethical clearance from the Research and Ethics Committee of MUHAS with reference number–MUHAS-REC-01-2023-1501. After ethical approval, the study team sought authorization to carry out the study from the Tanga Town Council and the Bombo Hospital administration. Before any participant could join the study, they had to go through the consenting process and sign the written informed consent after it was explained to them in

clear and plain language. A thumb-printed informed consent was obtained in the case of illiterate participants. Participants under the age of 18 were required to obtain permission from their parents or guardians. The study participants' confidentiality was ensured, and no participants' names were used in the questionnaire.

## Data management and analysis

Stata software version 15 was used for data analysis. The magnitude of adherence to ART among adolescents and young adult clients living with HIV was calculated from the eight-question one-month self-recall medication adherence scale tool. Participants who achieved an overall score of six or above (6–8) were categorized as adherent to ART, while those with a score below six (0–5) were categorized as non-adherent. Factors related to ART adherence among adolescents and young adults living with HIV were determined using logistic regression analyses. First, a bivariate logistic regression was conducted to examine the relationship between each independent variable and adherence to ART. Second, all independent variables associated with adherence to ART with a p-value < 0.2 in the bivariate regression were entered into the multiple logistic regression model. The results were reported as odds ratios (OR) and adjusted odds ratios (aOR) with 95% confidence intervals. All variables with small p values (P <0.05) were declared as statistically significant.

## Results

A total of 385 participants were interviewed during their attendance at the CTC clinic. **Table 1** summarizes the overall demographic characteristics of the study participants and adherence.

**Table 1. Socio-demographic characteristics of study participants (n = 385).**

| Variables | Categories | Overall n(%) |
|---|---|---|
| Age | 10–19 Years | 270 (70) |
| | 20–24 Years | 115 (30) |
| Sex | Female | 213 (55 |
| | Male | 172 (45) |
| Education level | Not attended school | 34 (9) |
| | Primary school | 147 (38) |
| | Secondary school | 194 (50) |
| | University/ college | 10 (3) |
| Marital status | Single | 368 (95.5) |
| | Married | 6 (1.6) |
| | Divorced | 5 (1.3) |
| | Co-habit | 5(1.3) |
| | Widow | 1 (0.3) |
| Occupation | Student | 222 (58) |
| | Employed | 15 (4) |
| | Self-employed | 66 (17) |
| | Unemployed | 82(21) |
| Location setting | Rural | 97 (25) |
| | Urban | 288 (75) |
| Family size | Less than 5 Family members | 125 (32) |
| | 5 family members or more. | 260 (68) |
| Adherence level | Adherent | 249 (64.7) |
| | Non-Adherent | 136 (35.3) |

The overall adherence was 64.7%. The participants' mean age was 17 years (SD±3.40), with a minimum age of 10 years and a maximum age of 24 years. Of all participants, 270 were adolescents and 115 were young adults. Most of the participants were female (55%), attended secondary school (50%), students (58%), single (95.5%), came from families with less than five members (55%), and lived in urban settings (75%).

### Factors associated with adherence to ART among participants

In multivariate analysis, participants with secondary education levels were 2.31 times more likely to adhere to ART compared to those with no education (p = 0.05). Household food insecurity was also associated with ART adherence. Those from households with moderate food insecurity were 67% less likely to adhere to ART compared to those with severe food insecurity (p <0.001).

  The availability of ART at the facility was also associated with adherence among adolescents and young adults. For example, those who always found ART available at the facility were 4.2 times more likely to adhere to the ART compared to their counterparts (p = 0.02). Moreover, side effects of medication were associated with ART adherence. Adolescents and young adults who experienced side effects were 39% less likely to adhere to ART compared to their counterparts (p = 0.04) (**Table 2**).

## Discussion

The study found that 35.3% of adolescents and young adult clients were non-adherent to ART. Factors associated with adherence included household food insecurity, low level of education, consistent availability of ART, and ART side effects. Addressing such unprecedented care and treatment challenges for HIV in this population calls for efforts to understand and mitigate these risk factors in the Tanga region as in other areas.

  More than one-third of adolescents and young adults on treatment were not adherent to life-saving ART. This surpasses the five percent threshold set by the World Health Organization [1]. Comparable findings from a separate study in Dar es Salaam, Tanzania which reported 37% [2]. Similar levels were found in Mwanza despite using viral and pill count methods [5]. Non-adherence levels to ART in some sub-Saharan African countries range from 13–20% [9], which are significantly lower compared to the findings in our study and the rest of the regions in Tanzania. The variations in ART adherence rates could stem from differences in data collection methods and the diverse approaches used to measure adherence [5,9,18,19]. For instance, methodologies such as pill count and pharmacy refill systems are commonly utilized but may carry the risk of overestimating adherence while recall methods might underestimate adherence levels [18,19]. Therefore, this study opted to employ an internationally validated method known for its high reliability [15].

  Like in previous studies [10,11,20], household food insecurity was associated with ART adherence in our study. Participants with food security were more likely to adhere to ART compared to those facing various degrees of food insecurity. This correlation aligns with findings in Tanzania [2], highlighting food insecurity as a barrier for young populations living with HIV in middle- and low-income countries, impeding optimal ART adherence [12]. Furthermore, a narrative review across Kenya, Uganda, South Africa, and other developed countries underscored the connection between food insecurity and poor adherence to ART, associating it with adverse HIV health outcomes such as lower CD4 cell counts and higher HIV viral loads in clients [10]. Inadequate nutrition resulting from food insecurity can weaken the immune system, thus compromising the effectiveness of ART and elevating the likelihood

**Table 2. Factors associated with adherence to ART among adolescents and young adults.**

| Variables | Freq (%) | Bivariate analysis | | | Multivariate analysis | | |
|---|---|---|---|---|---|---|---|
| | | cOR | 95% CI | P | aOR | 95% CI | P |
| **Education Level** | | | | | | | |
| Not attended school | 34 (9) | 1.00 | | | 1.00 | | |
| Primary school | 147(38) | 1.99 | 0.94–4.24 | 0.07 | 1.95 | 0.86–4.41 | 0.11 |
| Secondary school | 194 (50) | 2.51 | 1.20–5.26 | 0.02 | 2.31 | 1.02–5.23 | **0.05** |
| University/ college | 10 (3) | 1.13 | 0.27–4.61 | 0.87 | 0.85 | 0.18–3.94 | 0.83 |
| **Food insecurity** | | | | | | | |
| Food secure | 142 (37) | 1.00 | | | 1.00 | | |
| Mild insecure | 142 (37) | 0.76 | 0.45–1.27 | 0.30 | 0.76 | 1.42–1.36 | 0.35 |
| Moderate insecure | 77 (20) | 0.31 | 0.17–0.56 | 0.00 | 0.33 | 0.16–0.66 | **0.00** |
| Severe insecure | 24 (6) | 0.30 | 0.12–0.72 | 0.01 | 0.40 | 0.14–1.13 | 0.08 |
| **Wealth index** | | | | | | | |
| Lowest | 78 (20) | 1.00 | | | 1.00 | | |
| Lower | 76 (20) | 1.96 | 1.01–3.77 | 0.05 | 0.92 | 0.40–2.10 | 0.84 |
| Middle | 77 (20) | 1.88 | 0.98–3.60 | 0.06 | 0.96 | 0.46–2.00 | 0.91 |
| Higher | 77 (20) | 1.67 | 0.88–3.19 | 0.12 | 1.12 | 0.54–2.33 | 0.75 |
| Highest | 77 (20) | 2.12 | 1.10–4.10 | 0.03 | 1.42 | 0.69–2.96 | 0.34 |
| **Availability of ART** | | | | | | | |
| No | 17 (4) | 1.00 | | | 1.00 | | |
| Yes | 368 (96) | 4.72 | 1.63–13.71 | 0.00 | 4.20 | 1.29–13.72 | **0.02** |
| **Waiting time to pick ART** | | | | | | | |
| Less than 1 hour | 183 (48) | 1.00 | | | 1.00 | | |
| Between 1–5 hours | 152 (39) | 0.58 | 0.40–0.91 | 0.02 | 0.71 | 0.43–1.18 | 0.18 |
| More than 5 hours | 50 (13) | 0.42 | 0.22–0.80 | 0.01 | 0.64 | 0.31–1.31 | 0.22 |
| **Side effect of ART** | | | | | | | |
| No | 260 (68) | 1.00 | | | 1.00 | | |
| Yes | 125 (32) | 0.45 | 0.29–0.70 | 0.00 | 0.61 | 0.38–0.98 | **0.04** |
| **Frequency of daily dose** | | | | | | | |
| Once | 374 (97) | 1.00 | | | 1.00 | | |
| Twice | 11 (3) | 0.38 | 0.15–0.96 | 0.04 | 0.48 | 0.17–1.32 | 0.15 |

Key: cOR: Crude Odds ratio, aOR: Adjusted odds ratio, CI: Confidence interval.

of treatment failure. Moreover, the stress and anxiety stemming from food insecurity can adversely affect mental health, diminishing the motivation to adhere to ART.

Adherence to ART increased with higher education levels in this study, except for those with college-level education. Evidence suggests that adolescents and young adults with no formal education were more likely to be non-adherent compared to those with primary and secondary education [2,5]. Unexpectedly, individuals with university or college-level education demonstrated non-adherence contrary to studies in Tanzania [2] and Ethiopia [8]. It is expected that educated populations are more likely to understand health promotion and warnings compared to their counterparts [8,21]. They are more likely to look for explanations and search for further information beyond the counseling messages, therefore increasing the likelihood of adherence. Educated individuals may also be socially demographically advantageous in a better health decision state [2,8,21].

Availability and access to ART is an important factor behind adherence to medication [12]. The availability of ART at healthcare facilities demonstrated a strong association with

adherence among HIV-positive adolescents [12,21]. Similar evidence was reported in other Sub-Saharan African countries as reported in a review of evidence done between 2004 and 2016 [9]. This indicates that when ART is easily accessible, individuals are more inclined to adhere to their medication schedules. Therefore, ensuring the continual availability and accessibility of ART at healthcare facilities is vital for fostering optimal adherence to treatment and enhancing health outcomes among HIV-positive populations.

Side effects are a limitation to adherence among people living with HIV and AIDS. This was also demonstrated among adolescents and young adults in this study [12]. A narrative review conducted in sub-Saharan African countries reported that ART side effects hinder ART adherence among children and young adolescents [12]. Furthermore, a 2022 retrospective study in Dar es Salaam found that ART side effects were associated with poor adherence to ART [2]. Furthermore, the guidelines for HIV management in adolescents link side effects and inadequate adherence to ART. It highlights the importance of managing side effects effectively and providing adequate support to adolescents and young adults on ART [8,11]. ART counseling and investigating side effects during all clinic visits may help mitigate early signs and complaints of side effects [5].

The study was limited by its reliance on self-reported measures to collect data on ART adherence and related factors, which introduces the potential for recall bias. To address this limitation, clear and specific questions were used during data collection. Second, the study was done only at the Tanga Regional Referral Hospital. It did not include other levels such as district hospitals, health centers and dispensaries. Despite these limitations, this study fills the data void in adherence and factors associated with it among the vulnerable population in the Tanga region. Evidence presented in this study will strengthen efforts towards the fast-track targets.

In conclusion, the current study demonstrated suboptimal ART adherence among adolescents and young adults living with HIV and attending care and treatment facilities in Tanga region. More than a third of adolescents and young adults living with HIV do not adhere to this effective treatment. This level is far beyond the recommended WHO recommendation that may be on track to reach the 95-95-95 elimination targets. Factors such as food insecurity, lack of formal education, side effects, and ART shortages were associated with lower adherence rates. Addressing this unprecedented level of ART adherence, targeted interventions among adolescents and young adolescents are necessary to meet the targets. These include, improved patient education and support for adolescents and young adults with lower education levels; prioritizing interventions to address food insecurity, especially in such a region with adequate food production; ensuring consistent ART availability in care and treatment facilities at all times; and investigating, counseling and enhancing side effects management by healthcare providers, with a focus on routine monitoring and proactive support to improve overall adherence rates.

## Acknowledgments

The authors express gratitude to the study participants, research assistants, as well as the districts and regional administrative office, and Tanga Regional Referral Hospital for their invaluable contributions that have made this study successful.

## Author Contributions

**Conceptualization:** Sophia Kamote.

**Data curation:** Sophia Kamote.

**Formal analysis:** Sophia Kamote.

**Methodology:** Sophia Kamote.

**Supervision:** Novatus Apolinary Tesha, Bruno F. Sunguya.

**Writing – original draft:** Sophia Kamote.

**Writing – review & editing:** Sophia Kamote, Novatus Apolinary Tesha, Bruno F. Sunguya.

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
