## [Decision Letter · Decision Letter 0]

29 Jan 2024

PONE-D-23-39267Adherence to antiretroviral therapy among HIV-positive adolescents and young adult patients attending clinic at Bombo Hospital in Tanga region-TanzaniaPLOS ONE

Dear Dr. Tesha,

Thank you for submitting your manuscript to PLOS ONE. After careful consideration, we feel that it has merit but does not fully meet PLOS ONE’s publication criteria as it currently stands. Therefore, we invite you to submit a revised version of the manuscript that addresses the points raised during the review process.

Please submit your revised manuscript by Mar 14 2024 11:59PM. If you will need more time than this to complete your revisions, please reply to this message or contact the journal office at plosone@plos.org. Please include the following items when submitting your revised manuscript:

We look forward to receiving your revised manuscript.

Kind regards,

Werku Etafa

Academic Editor

PLOS ONE

Journal Requirements:

3. In the online submission form, you indicated that "Data will be available upon request. The authors will contact the ethical board to abide with the country policy on sharing data internationally."

4. We note you have included a table to which you do not refer in the text of your manuscript. Please ensure that you refer to Table 2 in your text; if accepted, production will need this reference to link the reader to the Table.

Additional Editor Comments:

**ACADEMIC EDITOR: **

Please provide your justifications for the critical questions raise by the reviewers.Pay attention to your English language usagImprove your discusion 

Reviewers' comments:

Reviewer's Responses to Questions

**Comments to the Author**

1. Is the manuscript technically sound, and do the data support the conclusions?

Reviewer #1: No

Reviewer #2: Yes

Reviewer #3: Yes

2. Has the statistical analysis been performed appropriately and rigorously? 

Reviewer #1: No

Reviewer #2: No

Reviewer #3: Yes

3. Have the authors made all data underlying the findings in their manuscript fully available?

Reviewer #1: No

Reviewer #2: Yes

Reviewer #3: Yes

4. Is the manuscript presented in an intelligible fashion and written in standard English?

Reviewer #1: Yes

Reviewer #2: No

Reviewer #3: Yes

5. Review Comments to the Author

Reviewer #1: Comments

Title: Adherence to antiretroviral therapy among HIV-positive adolescents and

young adult patients attending clinic at Bombo Hospital in Tanga region-Tanzania

1. Please reason out why you adhered to this age group?

Abstract

1. Background:

Why couldn't you consider the magnitude of poor ART drug adherence throughout the world or

if it is a public concern, rather than describing ART strength?

The readers should know what you mean by adherence to ART and its importance in adolescence

and young adulthood.

Please try to state your abstract by what is Adherence to ART, its magnitude, cause/risk factor,

the consequence of poor adherence, and conclude with your objective.

2. Methodology

Needs major revision

Please clearly describe here the statistical software used for data entering and the way

you generate statistically significant variables to your outcome variable.

3. Introduction

Please focus on Adherence to ART drug broadly (your objective is to assess the level of

Adherence to ART drug)

4. Methods;

Better to write study design and setting in one paragraph rather than separately.

Study area; must talk about the number and location of facilities in the study area and

number of facilities providing ART service. But you are talking here about the prevalence

of HIV, viral load which is absolutely not the right place to write.

How many referral hospitals are there in Tanga city?

What do you think of those health centers providing ART services and adolescent and

young adults attending ART treatment at health centers of Tanga City.

Study populations; What is the importance of stating the finding of others here? It's

unclear, and needs major revisions.

Sample size; detailed sample size calculations are needed considering estimated

prevalence of Adherence to ART drugs and factors significantly associated with

Adherence to ART drugs.

Data processing and management:

Initially, the process in this part is unclear.

How do you code, recode and compute the scale/items and dichotomized in-to Adherent

and Not adherent?

What is your baseline criteria to categorize in-to six or above and below?

Are you interested in logistic regression for only independent variables? What about for

dependent variables?

Is it bivariable or bivariate/multivariable or multivariate?

What did you mean by the results were reported as odds ratio and adjusted odds ratio?

Still you are adhered to p<0.2, how did you declare statistically significant variables?

Results:

Totally your result part needs major revisions

Please use scientific way of writing

Why are you writing the dependent variable in the sociodemographic characteristics?

Is it the right place to see the association and p-value in the sociodemographic

characteristics?

Age classifications?

Totally your regression table isWRONG

Reviewer #2: Thank you for giving me the chance to review this paper.

Some of the concerns are included below.

General

The paper needs language and grammar editing.

Abstract

Remove unnecessary abbreviations in the abstract.

Your conclusion needs revision; it would be better if you added your recommendation rather than repeating your findings.

Methods

It would be better if you added the health services related to the problem to the study area.

Why have you only considered a single institute as your study area?

Sample size determination

The author reported that he used 84% of the adherence level of the previous study, but it is not clear how it could be 384. Please clarify the procedure you used to calculate the sample size because I cannot obtain the number you mentioned using the single population proportion formula. In your description, you presented 385 data points. How could it be???

Data Processing and Analysis

The author said bivariable and multivariate analyses were carried out to check for significant variables. But you did not test the confounding factors, and you also didn’t check the goodness of fit of the model. Why?

Results

From the beginning, what is the significance of this study? What makes it different from the previous studies conducted in different areas of the country? Let me hear your new findings.

Look at your descriptive part; it should be modified to be more attractive for readers.

Discussion

Your justification is not strong enough; there are many studies conducted on your problem that are expected to be discussed with those findings. Rewrite it

Generally, the discussion part needs strong justifications.

References

Some of your references are outdated; please revise them.

Reviewer #3: Dear author,

I am pleased for your contribution in the study"Adherence to antiretroviral therapy among HIV-positive adolescents and young adult patients attending clinic at Bombo Hospital in Tanga region-Tanzania". I have comments apprehended below concerning your manuscript.

1. Summary of the research and your overall impression

When we say "Adherence to antiretroviral therapy among HIV-positive adolescents and young adult patients attending clinic" we are referring to the extent to which HIV-positive adolescents and young adults follow the prescribed treatment regimen for antiretroviral therapy (ART) at the clinic. This includes taking the medication as directed, following the recommended dosages, and adhering to the treatment schedule. Adherence to ART is essential for managing HIV infection and preventing disease progression. It also plays a crucial role in reducing the risk of transmission and improving the overall health outcomes of HIV-positive individuals. The study of adherence to ART among this specific demographic group aims to identify factors that influence adherence and develop interventions to improve treatment compliance.

1.1 Strengths of the manuscript

The manuscript's strength lies in its clear and precise writing, sentence coherence, and is focused on critical issues related to antiretroviral therapy (ART). The Authors have sufficiently reviewed relevant literature regarding adherence to ART among adolescents and young adults. The manuscript addresses an important component of adherence to ART among a vulnerable and understudied patient group. Therefore, the study's findings are expected to provide valuable insights into the barriers and facilitators of adherence in this population. This positions the manuscript as a significant contribution to the field of HIV care and management.

1.2 Weaknesses of the manuscript

The Authors did not use any figures to show concepts, procedures, or results.

2. Discussion of specific areas for improvement

2.1 Major issues

Sample size determination

You didn't calculate a sample size for your secondary objective (for factors) to check and take the large sample size.

The prevalence of adherence to ART was 84%. Therefore, you were supposed to use only 2% of margin of error to increase the number of your sample size.

2.2 Minor issues

In the abstract part

Please write the exposing and preventing factors separately. It seems a repetition of the main result. Therefore, mention only the names of variables significantly associated with your dependent variable. (Like A,B,C,and D were significantly associated.)

Better to write 'moderate food insecurity with (AOR 0.33 (95%CI 0.16-0.66)) and those experiencing ART side effects with AOR 0.61, 95%CI 0.38-0.98) were LESS LIKELY to adhere…..

X and Y (CI, AOR) were MORE LIKELY to ….

Please review the punctuation marks

Under introduction

Please, add more information about the solutions that have been tried in terms of successes and failures with only one paragraph.

Add gaps in the previous studies from what you reviewed.

Sample size determination

You did not mention the number of your total population. If your population is less than ten thousand, you could use a correction formula.

Under result

Be consistent while you write your paper. For instance, the prevalence of adherence to ART in the abstract part and the result part is not consistent (64.7 and 65%).

Table 2 Factors associated with adherence to ART

A frequency table should be included that shows the percentage of the adherent and non-adherent groups to see and check classification bias. Even it helps to check the Odds ratio manually.

Under discussion part

Your manuscript says that the differences could be owing to methods of data extraction methods risking under-estimation. It will be better if you clarify the tool differences (extraction methods) between yours and the others in a positive manner.

Reference related

There are some outdated references like Abrams E, Ammann A, Anderson M, Bernstein L, Baker C. Guidelines for the use of antiretroviral agents in pediatric HIV Infection January 7, 2000. HIV Clin Trials. 2000;1(3):58–99. 30

6. PLOS authors have the option to publish the peer review history of their article (what does this mean?). If published, this will include your full peer review and any attached files.

Reviewer #1: No

Reviewer #2: No

Reviewer #3: No

---

## [Author Response · Author response to Decision Letter 0]

1 Jul 2024

The responses to reviewers comments have been put in a matrix with a responses and page number where in the track changed version the changes occured.

---

## [Decision Letter · Decision Letter 1]

6 Aug 2024

PONE-D-23-39267R1Factors associated with adherence to antiretroviral therapy among HIV-positive adolescents and young adult patients attending clinic at Bombo Hospital in Tanga region-Tanzania.PLOS ONE

Dear Dr. Tesha,

Thank you for submitting your manuscript to PLOS ONE. After careful consideration, we feel that it has merit but does not fully meet PLOS ONE’s publication criteria as it currently stands. Therefore, we invite you to submit a revised version of the manuscript that addresses the points raised during the review process.

We look forward to receiving your revised manuscript.

Kind regards,

Werku Etafa

Academic Editor

PLOS ONE

Journal Requirements:

Reviewers' comments:

Reviewer's Responses to Questions

**Comments to the Author**

1. If the authors have adequately addressed your comments raised in a previous round of review and you feel that this manuscript is now acceptable for publication, you may indicate that here to bypass the “Comments to the Author” section, enter your conflict of interest statement in the “Confidential to Editor” section, and submit your "Accept" recommendation.

Reviewer #1: (No Response)

2. Is the manuscript technically sound, and do the data support the conclusions?

Reviewer #1: Yes

3. Has the statistical analysis been performed appropriately and rigorously? 

Reviewer #1: Yes

4. Have the authors made all data underlying the findings in their manuscript fully available?

Reviewer #1: (No Response)

5. Is the manuscript presented in an intelligible fashion and written in standard English?

Reviewer #1: No

6. Review Comments to the Author

Reviewer #1: (No Response)

7. PLOS authors have the option to publish the peer review history of their article (what does this mean?). If published, this will include your full peer review and any attached files.

Reviewer #1: No

---

## [Author Response · Author response to Decision Letter 1]

22 Aug 2024

The responses have been attached with a response matrix

---

## [Decision Letter · Decision Letter 2]

8 Dec 2024

Factors associated with adherence to antiretroviral therapy among HIV-positive adolescents and young adult patients attending HIV care and treatment clinic at Bombo Hospital in Tanga region-Tanzania

PONE-D-23-39267R2

Dear Dr. Tesha,

We’re pleased to inform you that your manuscript has been judged scientifically suitable for publication and will be formally accepted for publication once it meets all outstanding technical requirements.

Kind regards,

Joel Msafiri Francis, MD, MS, PhD

Academic Editor

PLOS ONE

Additional Editor Comments (optional):

Reviewers' comments:

Reviewer's Responses to Questions

**Comments to the Author**

1. If the authors have adequately addressed your comments raised in a previous round of review and you feel that this manuscript is now acceptable for publication, you may indicate that here to bypass the “Comments to the Author” section, enter your conflict of interest statement in the “Confidential to Editor” section, and submit your "Accept" recommendation.

Reviewer #1: All comments have been addressed

Reviewer #4: All comments have been addressed

2. Is the manuscript technically sound, and do the data support the conclusions?

Reviewer #1: Yes

Reviewer #4: Yes

3. Has the statistical analysis been performed appropriately and rigorously? 

Reviewer #1: Yes

Reviewer #4: Yes

4. Have the authors made all data underlying the findings in their manuscript fully available?

Reviewer #1: Yes

Reviewer #4: Yes

5. Is the manuscript presented in an intelligible fashion and written in standard English?

Reviewer #1: Yes

Reviewer #4: Yes

6. Review Comments to the Author

Reviewer #1: (No Response)

Reviewer #4: Title and Abstract

The title and abstract are well-written and effectively summarize the study's objectives, methodology, and key findings.

However, there is a grammatical error on line 34, page 2, where the sentence reads, “…more 4.2 times more likely…”. The repetition of the word "more" needs to be corrected to improve clarity and readability.

Introduction

The introduction is well-constructed, providing a clear and relevant background to the study's focus. It effectively highlights the importance of addressing adherence to antiretroviral therapy among HIV-positive adolescents and young adults.

Methods

The methods section is comprehensive and addresses all the necessary elements as outlined in the STROBE checklist. The design, sampling, data collection, and analysis approaches are clearly described, enabling replication and enhancing the credibility of the findings.

Results

The results are clearly presented and logically structured.

There is a minor formatting error in Table 1, line 189, page 10: a closing bracket is missing after the percent score for the female gender (55%). Please ensure the bracket is included to maintain consistency in table formatting.

Discussion

The discussion section is well-written, providing a thoughtful interpretation of the findings in the context of existing literature. The authors appropriately discuss the implications of their findings.

7. PLOS authors have the option to publish the peer review history of their article (what does this mean?). If published, this will include your full peer review and any attached files.

Reviewer #1: No

Reviewer #4: No

---

## [Editor Report · Acceptance letter]

22 Dec 2024

PONE-D-23-39267R2 

PLOS ONE

Dear Dr. Tesha, 

I'm pleased to inform you that your manuscript has been deemed suitable for publication in PLOS ONE. Congratulations! Your manuscript is now being handed over to our production team.

Kind regards, 

on behalf of

Prof. Joel Msafiri Francis 

Academic Editor

PLOS ONE